# A 100 bp GAGA motif-containing sequence in *AGAMOUS* second intron is able to suppress the activity of CaMV35S enhancer in vegetative tissues

**Ruochen Liu, Xiuping Zou, You Wang, Qin Long, Yan Pei***

Chongqing Key Laboratory of Application and Safety Control of Genetically Modified Crops; Biotechnology Research Center, Southwest University, Beibei, Chongqing, P. R. China

* peiyan3@swu.edu.cn

**Data Availability Statement:** All relevant data are within the manuscript and its Supporting Information files.

## Abstract

Flower-specific promoters enable genetic manipulation of floral organs to improve crop yield and quality without affecting vegetative growth. However, the identification of strong tissue-specific promoters is a challenge. In addition, information on *cis* elements that is able to repress gene expression in vegetative tissues remains limited. Here, we report that fusing a 35S enhancer to the stamen- and carpel-specific NtAGIP1 promoter derived from the tobacco *AGAMOUS* second intron (AGI) can significantly increase the promoter activity. Interestingly, although the activity of the new promoter extends to sepals and pedicles, it does not cross the boundary of the reproductive organs. Serial deletion of the AGI and chromatin immunoprecipitation (ChIP) assay reveal a 100-bp fragment that contains a conserved GAGA factor binding motif contributes to the flower specificity by mediating histone H3 lysine 27 trimethylation (H3K27me3) modification of the promoter. Furthermore, this fragment shows significant suppressive effect on the activity of the 35S enhancer in vegetative tissues, consequently, resulting in a significant increase of the activity of 35S enhancer: AGI chimeric promoter without sacrifice of its specificity in inflorescence.

## Introduction

To generate desired transgenic plants, both transgenes that confer the desired traits and promoters that instruct expression of transgene in targeted tissues are needed [1,2]. Constitutive promoters, such as cauliflower mosaic virus 35S (CaMV35S) [3], maize ubiquitin [4], and rice actin promoters [5] have been widely used in both basic and applied studies. Among these, the CaMV35S promoter is the most frequently used one [1,6,7]. However, in many cases high and universal expression of target genes is nutrient and energy consuming. Moreover, constitutive expression of some genes can produce negative impacts on plant growth or agronomic performance [8–10].

Tissue-specific promoters allow the expression of targeted genes only in specific tissues [11,12]. Reproductive tissues (*e.g.*, flower and seed) are frequent targets for transgene

**Funding:** This research was supported by the Chinese Ministry of Science and Technology of China (Grant 2016YFD0100505). The funders had no role in study design, data collection and analysis, decision to publish, or preparation of the manuscript.

**Competing interests:** The authors have declared that no competing interests exist.

expression to improve yield or commercial value of plants [13–15]. Thus far, many flower specific promoters have been isolated and characterized. However, such promoters often have some drawbacks for applications, *e.g.*, low activity or non-specific leakage [1,10]. To increase expression of target genes or to avoid homology-based gene silencing, a variety of strategies have been proposed, such as dCas9-based gene activation tool [16], codon optimization [17], synthetic promoters [2]. Among them, chimeric or synthetic promoter capable of directing gene expression with desired strength in right tissue or right developmental stage of plants is a simple one. Enhancer is a *cis*-acting DNA sequence that can stimulate transcription from proximal promoters in a distance- and orientation-independent manner [18,19]. To increase the activity of tissue-specific promoters, a simple and straightforward strategy is fusing an enhancer to the promoter.

AGAMOUS (*AG*) is a MADS-box gene that acts in the inner two whorls of *Arabidopsis thaliana* flowers to specify stamens and carpels [20,21]. The expression of *AG* is strictly confined to carpel and stamen primordia and tissues [22,23]. It has been shown that the second intron/enhancer (AGI) of *AG* is responsible for conferring the tissue specificity of the gene [24]. In transgenic Arabidopsis, *AG* native promoter alone drives GUS expression in both vegetative and floral tissues. With the presence of the second intron/enhancer, GUS expression is specifically localized to carpels and stamens [25]. Furthermore, artificial promoter generated by fusing AtAGI to the minimal 35S promoter (AtAGIP) is sufficient to drive gene expression precisely in carpels and stamens [26,27]. Yang and colleagues (2010) isolated two similar *AG* second intron/enhancers, NtAGI-1 and NtAGI-2, from tetraploid tobacco (*Nicotiana tabacum*). Fusing them to the minimal 35S promoter, they generated NtAGIP1 and NtAGIP2 promoters. Like AtAGIP, the two promoters are able to drive carpel- and stamen-specific expression without any leaky activity in vegetative tissues [28].

To increase the activity of NtAGIP1, in this study, we added a 35S enhancer to the promoter. The enhancer significantly increased the promoter activity. Interestingly, although extended to sepals and pedicles, the activity of the new promoter was still confined to the inflorescence. We showed that the -2835 to -2735 region of NtAGI-1 which contains a conserved GAGA factor binding motif can suppress the activity of 35S enhancer in vegetative tissues by mediating histone H3 lysine 27 trimethylation (H3K27me3) modification of the promoter. Our results provide useful information for the improvement of tissue-specific promoters, and the resultant 35SNtAGIP1 promoter can be used in transgenic plants when strong expression of target gene in flowers is required.

## Materials and methods

### Plant material and growth condition

Common tobacco, *Nicotiana tabacum* cv. *Xanthi*, was used in this study. Tobacco plants were grown on soil in a greenhouse under natural day length conditions. To generate sterilized plantlets for transformation, seeds were first surface-sterilized, and then germinated on MS solid medium supplemented with 30% sucrose. The seed sterilization process consists of 75% ethanol 1 min; 1% sodium hypochlorite (NaOCl) 15 min; wash 6–8 times with sterilized water. Plated seeds were germinated and grown in a growth chamber under a 16 h light and 8 h dark photoperiod at 28˚C.

### Plasmid construction

The plasmid pBI121 [29] that contain *CaMV35S*::*GUS* was used as backbone to construct vectors used for plant transformation in this study. The second intron of *NtAG-1* gene was fused with a 45 bp minimal 35S promoter to create a functional NtAGIP1 promoter [28]. *NtAGIP1*::

*GUS* vector was constructed by replacing the CaMV35S promoter of pBI121 using *Sal*I and *Bam*HI restriction sites. For constructing *35SNtAGIP1::GUS* vector, the 35S enhancer (-396 to -46) was added at the 5' end of NtAGIP1 using *Sal*I and *Spe*I restriction sites that had been introduced during the construction of *NtAGIP1::GUS* vector. For generating *Promoterless::GUS* vector, the CaMV35S promoter of pBI121 was deleted by *Sal*I and *Bam*HI restriction enzymes.

For serial deletions of the NtAGI-1, shortened NtAGIP1 sequences were PCR amplified with the PrimeSTAR Max DNA Polymerase (Takara) using primers listed in the S1 Table. Each shortened version of NtAGIP1 had *Spe*I and *Bam*HI restriction sites. Then, the fragments are cloned into the *Spe*I/*Bam*HI-digested *35SNtAGIP1::GUS* plasmid, resulting in replacing the full length NtAGIP1 with the shortened NtAGIP1 sequences.

To test the inhibitory effect of the 100-bp repressive fragment of NtAGI-1, the three promoters, 35SRF, 35SUC and 35SDC, were synthesized directly, and then fused to upstream of *GUS* gene in the pBI121 plasmid using *Sal*I and *Bam*HI restriction sites, generating *35SRF::GUS*, *35SUC::GUS* and *35SDC::GUS* vectors.

## Plant transformation

The resulting plasmids were introduced into *Agrobacterium tumefaciens* (LBA4404) by electroporation. Leaf disc transformation of tobacco using the *Agrobacterium tumefaciens* was performed as previously described [30]. Briefly, bacteria grown in YEB for 20 h were harvested by centrifugation and resuspended to an $OD_{600}$ of 0.8–1.0 in MS medium supplemented with 10 g/L sucrose. Aseptic leaves were cut into pieces (around 6 mm in diameter) and the leaf discs were inoculated with the bacterial suspension for 30 min, blotted on sterile filter paper, and then transferred to co-cultivation medium (MS salt containing vitamins, 2.0 mg/L 6-BA, 0.5 mg/L NAA, 10 g/L sucrose, 2.0 g/L Gelrite, pH 5.4). After maintained at 25°C in the dark for 2 days, the transformed leaf discs were transferred onto selection medium I (MS salt containing vitamins, 2.0 mg/L 6-BA, 0.5 mg/L NAA, 200 mg/L cephalosporin, 50 mg/L kanamycin, 30 g/L sucrose, 2.0 g/L Gelrite, pH 5.8). When visible calli appeared on explants, the cultures were transferred to selection medium II (MS salt containing vitamins, 200 mg/L cephalosporin, 50 mg/L kanamycin, 30 g/L sucrose 2.0 g/L Gelrite, pH 5.8) for shoots production and root elongation. Kanamycin-resistant candidate transgenic plants were verified by PCR amplifying the *NptII* gene with 2×Taq Master Mix (Novoprotein, China), and then were transplanted and grown in the greenhouse.

## Histochemical detection and microscopy

Histochemical assay of GUS activity was performed as described previously [29]. Detached or hand-sectioned tissues were incubated in GUS staining solution (1 mM X-Gluc in 100 mM sodium phosphate (pH 7.0), 0.5 mM $K_4[Fe(CN)_6]$, 0.5 mM $K_3[Fe(CN)_6]$, and 0.1% Triton X-100, 10 mM EDTA-$Na_2$) at 37°C in the dark overnight. Then, the tissues were bleached with 95% ethanol before photographing. Microscopic observation was performed with OLYMPUS-MVX10 microscope (Olympus). Images were captured with Image Pro-Plus software (Media Cybernetics).

## Fluorometric assays of GUS activity

Fluorometric assay of GUS activity was conducted as described by Jefferson et al [29]. Four lines with strong GUS staining signal of *NtAGIP1::GUS* and *35SNtAGIP1::GUS* transgenic tobaccos, and all lines of *35SRF::GUS*, *35SUC::GUS* and *35SDC::GUS* transgenic tobaccos were used for the assay. Tissues were ground in liquid nitrogen with mortar and pestle and homogenized in extracting buffer (50 mmol/L sodium phosphate buffer containing 100 mg/mL PVPP and 10 mmol/L β-mercaptoethanol, PH 7.0). After incubation on ice for 1 h, the extraction

mixture was centrifuged at 15000×g for 10 min. The supernatant was used for GUS activity assay in which 20 μL supernatant was mixed with 50 μL assay buffer containing 1.4 mM 4-methylumbelliferone (4-MU) (Sigma). The reaction was incubated at 37˚C for 30 min and then stopped with 130 μL stop solution containing 200 mM sodium carbonate. The liberation of 4-MU was detected by measuring the fluorescence on a microplate reader (TECAN infinite M200 PRO) with excitation at 365 nm and emission at 455 nm and standardized against known concentrations of 4-MU diluted in the buffer. The protein concentration was determined by the Bradford assay [31], using bovine serum albumin as the standard. GUS activity was calculated as nmol 4-MU/min/mg protein and each test was performed with three biological replicates.

### H3K27me3 chromatin immunoprecipitation (ChIP)

Tobacco leaves (0.8–1.0 g fresh weight) were harvested and used for the H3K27me3 ChIP assay. ChIP was performed with EpiQuik^TM Plant ChIP Kit (Epigentek, http://www.epigentek.com) following the manufacturer's instructions. The anti-H3K27me3 antibody (Millipore, http://www.emdmillipore.com) was used for immunoprecipitation. 2.0 μL of immunoprecipitated DNA was used for qRT-PCR experiments. Primers used for the qRT-PCR detection are listed in the S2 Table. qRT-PCR was performed with 1×iQ™ SYBR Green Supermix (Bio-Rad) on CFX96TM Real-Time System (Bio-Rad). The thermal cycling consisted of an initial denaturation (94˚C, 3 min) followed by 40 cycles (94˚C, 20 s; 56˚C, 20 s; 72˚C, 30 s). ChIP assays were performed with three biological replicates.

### Statistical analysis

Data were statistically analyzed by Microsoft Excel. Means and standard deviations (SD) of values are shown. Statistical comparison was analyzed with two-tailed Student's *t*-test, and indicated by asterisks (*, $P < 0.001$) or NS (NS, not significant, $P > 0.05$).

## Results

### Adding 35S enhancer to NtAGIP1 broadens the tissue specificity from carpels and stamens to whole inflorescence

To increase the promoter activity, we added 35S enhancer fragment (-396 to -46) [32,33] to the 5 'end of NtAGIP1, and generated 35SNtAGIP1 promoter (Fig 1A). Then, NtAGIP1 and 35SNtAGIP1 were fused to the *GUS* coding region to create *NtAGIP1*::*GUS* and *35SNtAGIP1*::*GUS* fusions (Fig 1A). *CaMV35S*::*GUS* and *Promoterless*::*GUS* were used as positive and negative controls, respectively (Fig 1A). The four expressing vectors were used to generate transgenic tobaccos through *Agrobacterium*-mediated transformation using kanamycin resistant *NptII* gene as the selectable marker. Putative transformants were verified by PCR detection, and the positive transformants were selected for studies.

Histochemical analysis of GUS activity was performed to examine the expression pattern driven by the promoters (Fig 1B–1E). *CaMV35S*::*GUS* lines, the positive control, exhibited strong GUS staining in all vegetative tissues, including leave, stem, and root tissues. Consistent with previous report [28], GUS activity in *NtAGIP1*::*GUS* line was observed only in stamens and carpels. Interestingly, adding 35S enhancer to the 5' end of NtAGIP1, the activity of the promoter was observed not only in stamens and carpels, but also in sepals and pedicels (Fig 1B–1E), indicating that adding 35S enhancer broadens the tissue-specificity in carpels and stamens to the whole inflorescence. Nevertheless, no GUS signal was observed in the vegetative tissues including roots, stems and leaves of *35SNtAGIP1*::*GUS* lines.

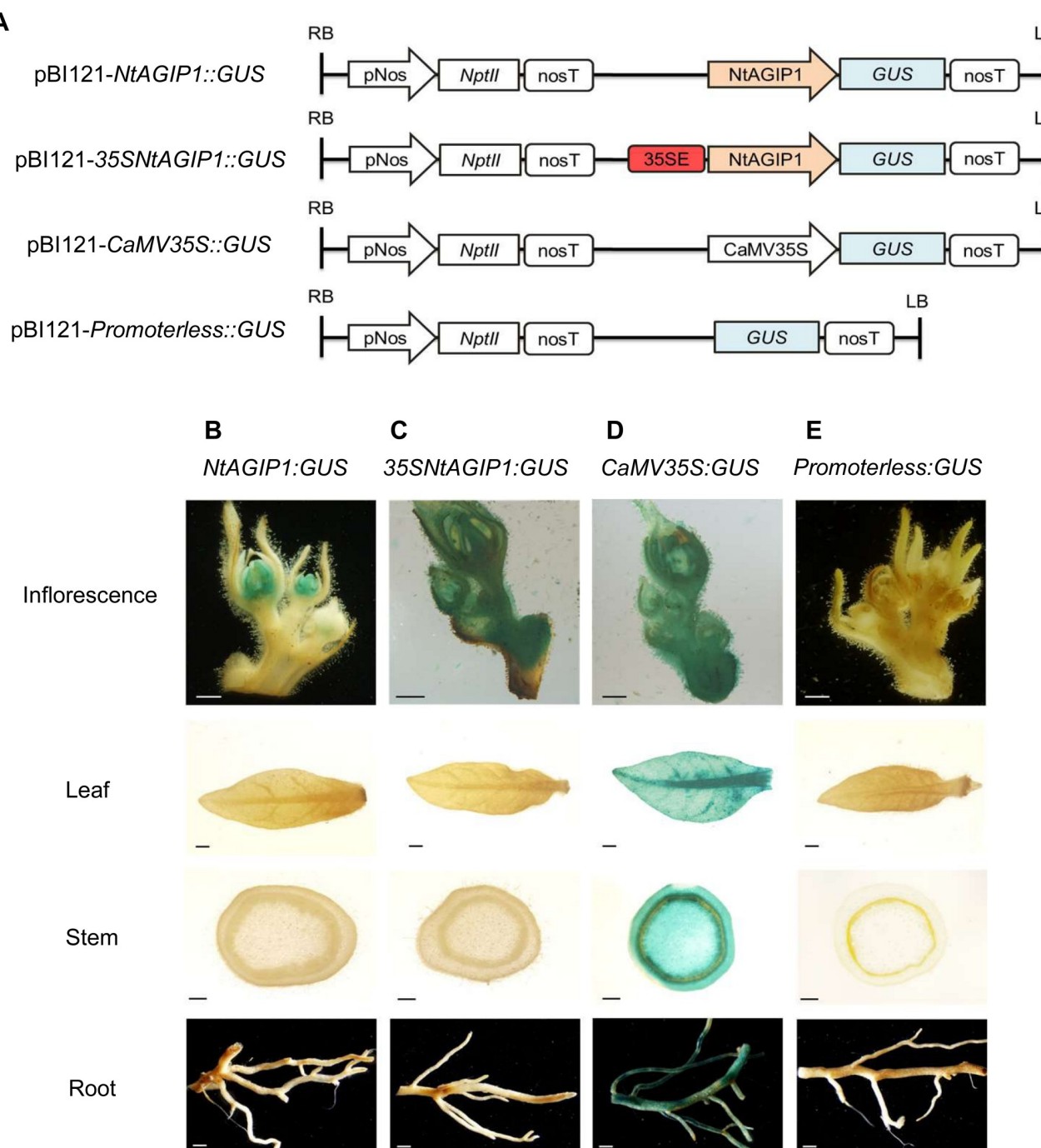

**Fig 1. Adding 35S enhancer broadens the tissue specificity of NtAGIP1 in inflorescence.** (**A**) Schematic diagram of constructs based on binary vector pBI121 for the synthetic promoter assays. (**B-E**) Expression patterns of *GUS* gene driven by different promoters in (A). Representative GUS patterns are shown for *NtAGIP1::GUS* (B), *35SNtAGIP1::GUS* (C), *CaMV35S::GUS* (D), and *Promoterless::GUS* (E) in transgenic reporter lines. Independent transgenic lines (n > 20 for each construct) were assayed for GUS expression in inflorescences, leaves, stems and roots. Scale bars, 1 mm.

## Adding 35S enhancer to NtAGIP1 increases the promoter activity

We noted that compared with NtAGIP1, adding the 35S enhancer also increased the activity of NtAGIP1 promoter. Based on the intensity of GUS staining, transgenic lines were classified

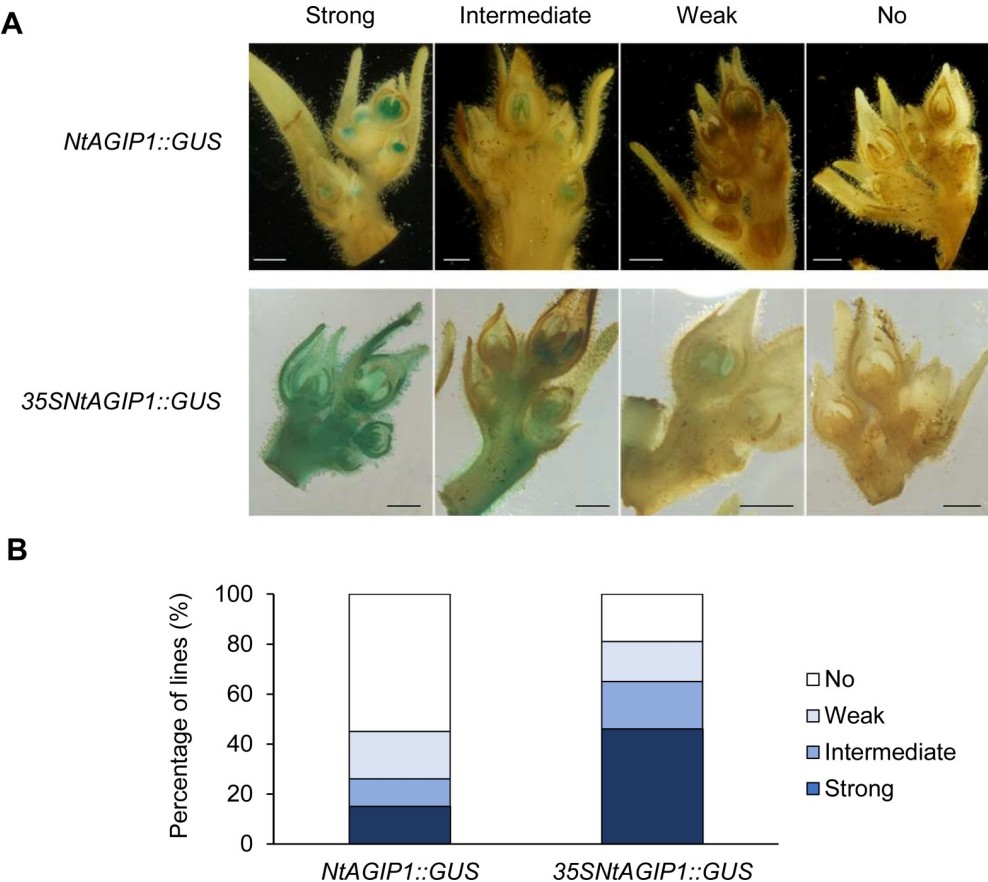

**Fig 2. Adding 35S enhancer increased the promoter activity. (A)** Examples of strong, weak, intermediate, and no GUS staining in inflorescence of *NtAGIP1*::*GUS* and *35SNtAGIP1*::*GUS* lines. Scale bars, 1 mm. (**B**) Statistics of lines showing strong, intermediate, weak, or no GUS staining in inflorescence of *NtAGIP1*::*GUS* and *35SNtAGIP1*::*GUS* transgenic tobaccos.

into four groups: strong, intermediate, weak, and no staining (Fig 2A). In *NtAGIP1*::*GUS* lines (n = 27) tested, 15%, 11%, 19%, and 55% lines displayed strong, intermediate, weak, and no staining respectively. While in *35SNtAGIP1*::*GUS* lines (n = 37), the proportion of strong, intermediate, weak, and no staining was 46%, 19%, 16%, and 19% respectively (Fig 2B). Quantification of GUS activity of the four strong expressing lines of each group confirmed that the gene expression in the inflorescence and inner whorls of flowers of *35SNtAGIP1*::*GUS* lines was much higher than that of *NtAGIP1*::*GUS* lines (Fig 3). The data indicate that adding the 35S enhancer can remarkably enhance the activity of NtAGIP1 promoter in inflorescent tissues.

## The -2835 to -2735 region of NtAGIP1 contributes to the suppression of vegetative expression

Now the question is why adding the 35S enhancer can enhance the activity of NtAGIP1 promoter but not impair its specificity in the reproductive organs. To answer it, a progressive deletion of full length NtAGIP1 (~4.2kb) promoter was performed. Each shortened sequence was fused with 35S enhancer at its 5' end and then linked to *GUS* reporter gene (Fig 4A). The resultant constructs were delivered into tobacco, respectively. GUS expression in leaves of

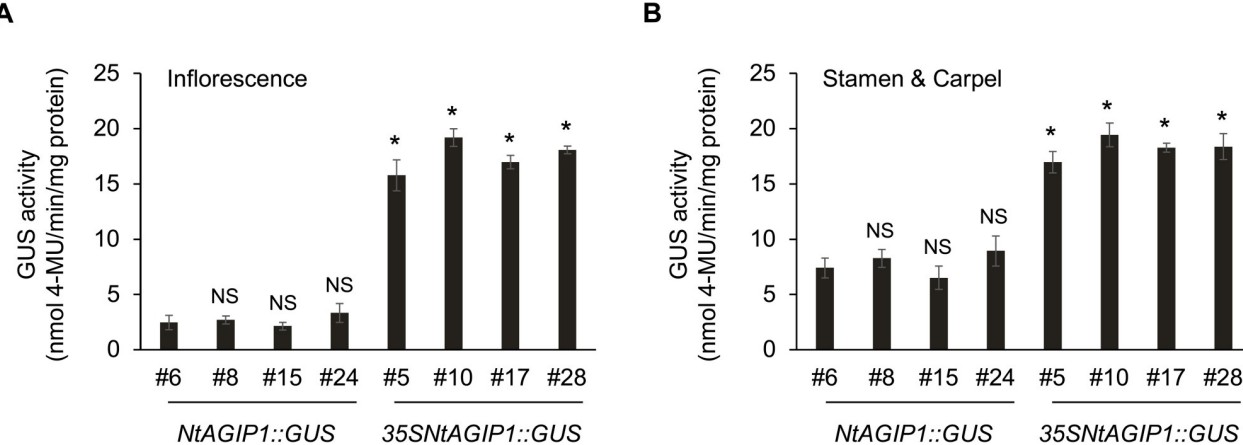

**Fig 3. Adding 35S enhancer enhanced the GUS activity in the inflorescence.** GUS activity in the inflorescence (**A**), and in inner whorls of the flower (**B**). Four lines of *NtAGIP1*::*GUS* and *35SNtAGIP1*::*GUS* transgenic tobaccos with strong GUS expression were selected randomly for GUS activity assays. Error bars represent standard deviations of three biological replicates. *, $P < 0.001$; NS, not significant ($P > 0.05$) relative to #6 line of *NtAGIP1*::*GUS* tobacco; two-tailed Student's *t*-test.

more than 30 transgenic lines for each construct was detected and the expression frequency was calculated (Fig 4A). When the deletion reached to the upstream -2835, almost no GUS expression was observed in leaves. However, when the deletion arrived at -2365 and thereafter, GUS expression was significantly detected in the populations. In -2365 construct, the GUS expression frequency was 36.3%, and further deletion had no more significant effect on the GUS expression frequency. These results suggested that putative repressive elements exist between -2835 and -2365. To characterize the location of potential repressive elements between -2835 and -2365, we conducted a new series of deletion by every 100 bp. When the deletion extended to -2735, GUS expression was observed in 9 out of 31 lines. Further deletion to -2635, -2535, and -2435 did not result in a significant increase of the GUS expression frequency (Fig 4B). These results firmly demonstrate that putative repressive elements locate in the 100-bp region between -2835 and -2735 of NtAGI-1.

## The -2835 to -2735 region of NtAGI-1 can repress the vegetative activity of 35S enhancer

To test the repressive role of the -2835 to -2735 fragment of NtAGI-1 in the vegetative expression, we fused the fragment between the 3' end of the 35S enhancer and 5' end of the 35S minimal promoter, and then generated 35SRF (Repressive Fragment) promoter (Fig 5A). Two 100-bp DNA fragments (-2934 to -2834 and -2736 to -2636) from up- and down-stream of the repressive fragment were used as controls to generate promoters 35SUC (Upstream Control) and 35SDC (Downstream Control), respectively (Fig 5A). The promoters were fused to *GUS* gene respectively, and then transgenic *35SRF*::*GUS*, *35SUC*::*GUS*, and *35SDC*::*GUS* tobaccos were generated. GUS staining showed that 86.7% (26/30) of *35SUC*::*GUS*, and 88.5% (23/26) of *35SDC*::*GUS* reporter lines showed obvious GUS expression in leaves. In contrast, 50.0% (17/34) of *35SRF*::*GUS* lines showed detectable GUS signal in leaves (Fig 5B). Noticeably, GUS activity in leaves of *35SRF*::*GUS* lines was significantly lower than ($P < 0.001$) that of *35SUC*::*GUS* and *35SDC*::*GUS* lines (Fig 5C). The data indicate that the -2835 to -2735 repressive fragment of NtAGI-1 has ability to repress activity of the constitutive enhancer in the vegetative tissue.

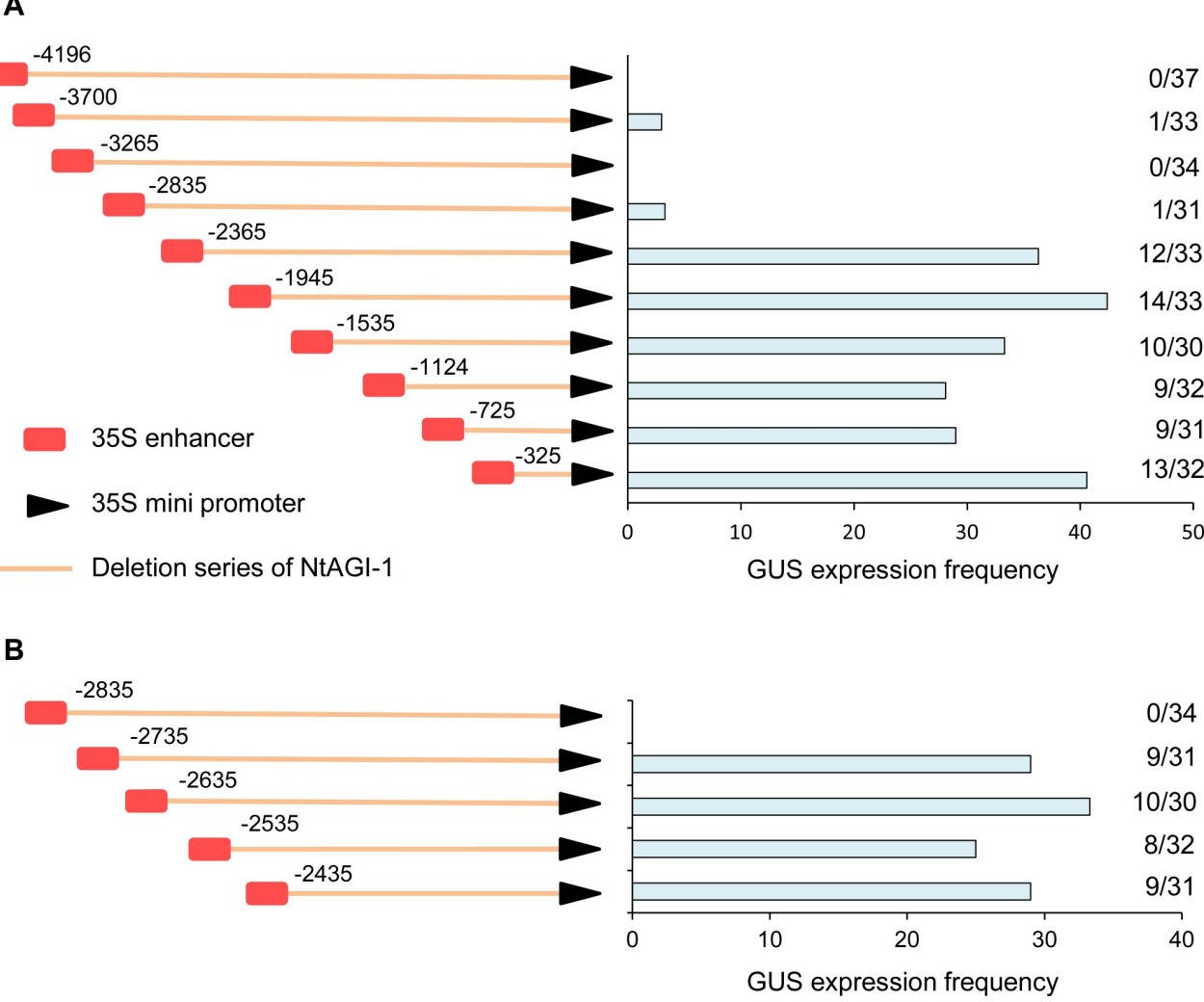

**Fig 4. Dissection analysis of the *NtAG* intron to dig potential repressive elements.** (**A**) Schematic diagram of constructs of serial deletions of NtAGIP1 (left), and GUS expression frequency in leaves of transgenic tobacco lines for each construct (right). (**B**) Further serial deletions of fragment between -2835 and -2365 for precise localization of potential repressive elements (left), and GUS expression frequency in leaves of transgenic tobacco lines for each construct (right).

## The -2835 to -2735 region of NtAGI-1contains GAGA factor binding motif

Bioinformatic analysis result showed that in the -2835 to -2735 region of NtAGI-1 there was a GAGA factor binding motif (Fig 6) that is involved in the recruitment of polycomb repressive complex 2 (PRC2) [34–36]. PRC2 is responsible for the repression of flower genes, including *AG*, in vegetative tissues. Interestingly, all *AG* homologs of the five Solanaceae plants, tobacco, tomato, potato, petunia and pepper, have a big second intron (Fig 7), in which the 100-bp repressive fragment of NtAGI-1 with GAGA factor binding motif is conserved among the species of Solanaceae family (Fig 6 and Table 1).

It has been known that PRC2-mediated repression of flower genes is resulted from H3K27me3 modification to form epigenetically stable silent chromatin state [37,38]. Our chromatin immunoprecipitation (ChIP) assay showed that in the wild-type tobacco the entire NtAGI-1 intron including the 100-bp repressive fragment was enriched for H3K27me3 modification (Fig 8A). In the GUS positive -2735 cells, H3K27me3 level of the chimeric promoter was

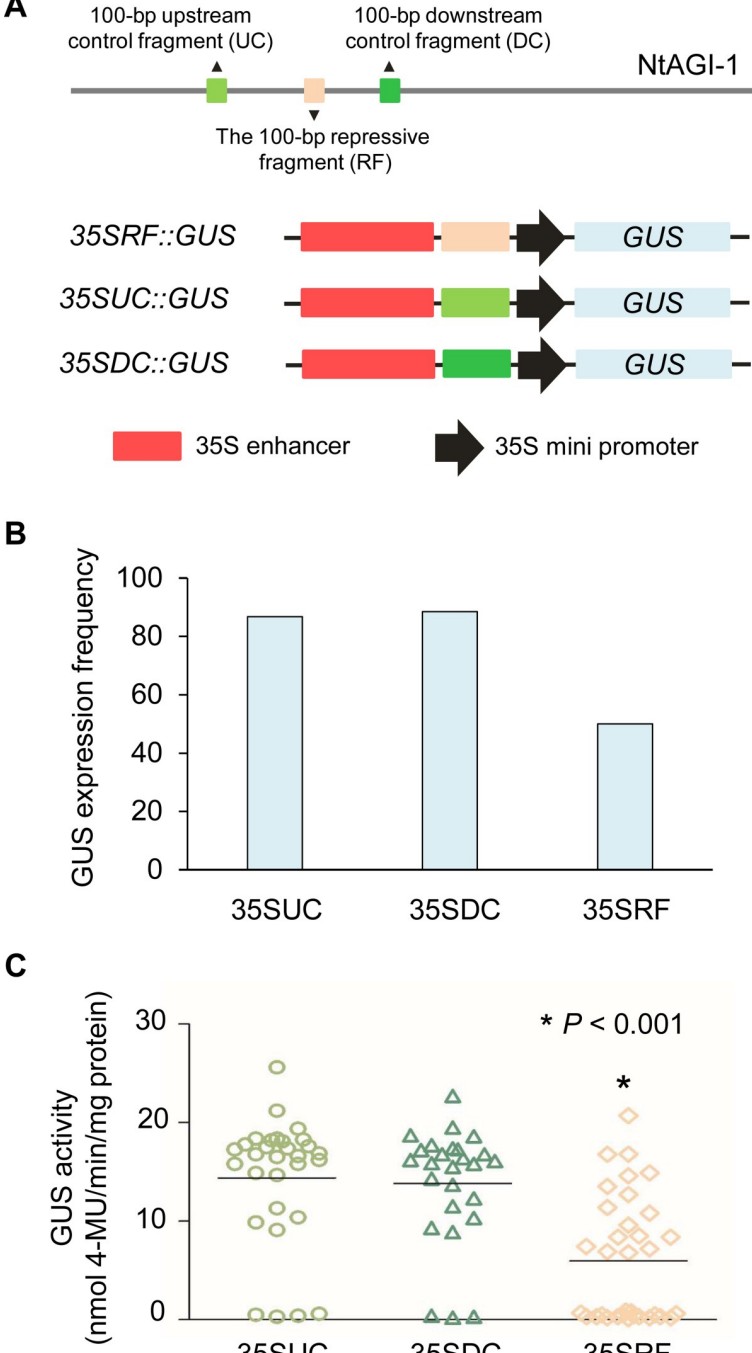

**Fig 5. The 100-bp repressive fragment of NtAGI-1 is able to suppress 35S enhancer activity in leaves.** (**A**) Schematic diagram of *GUS* reporter constructs. (**B**) GUS expression frequency driven by the chimeric promoter containing 35S enhancer and the 100-bp repressive fragment and by two control promoters. (**C**) Quantification of GUS activity in the three transgenic groups. Statistical significance was determined by two-tailed Student's *t*-test. The horizontal lines represent the medians.

much lower than that in the GUS negative -2835 cells (Fig 8B), indicating that the 100-bp region can mediate H3K27 tri-methylation to suppress the activity of the promoter. These results suggest the crucial role of GAGA motif in the 100-bp region for the vegetative repression.

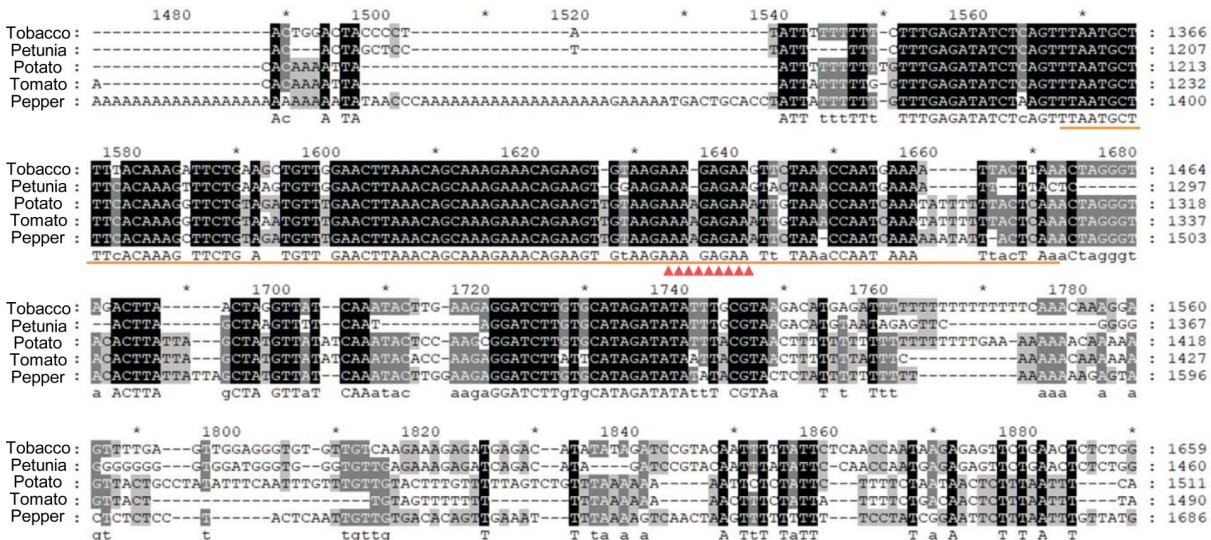

**Fig 6. The 100-bp repressive fragment (orange line) and GAGA motif (red triangles) of NtAGI-1 are conserved in the species of Solanaceae family.**

## Discussion

It is often a dilemma when design a strong tissue-specific promoter: increasing the strength of the promoter is frequently accompanied by the loss of tissue specificity. In this study, we show that the well-known strong enhancer (-396 to -46) of 35S promoter can be harnessed for the design of the chimeric flower-specific promoter by fusing it with the floral organ-specific promoter NtAGIP1. The strength of the promoter in the flower is significantly enhanced. Meanwhile, its activity is confined to the reproductive organs. We further demonstrate that a 100-bp

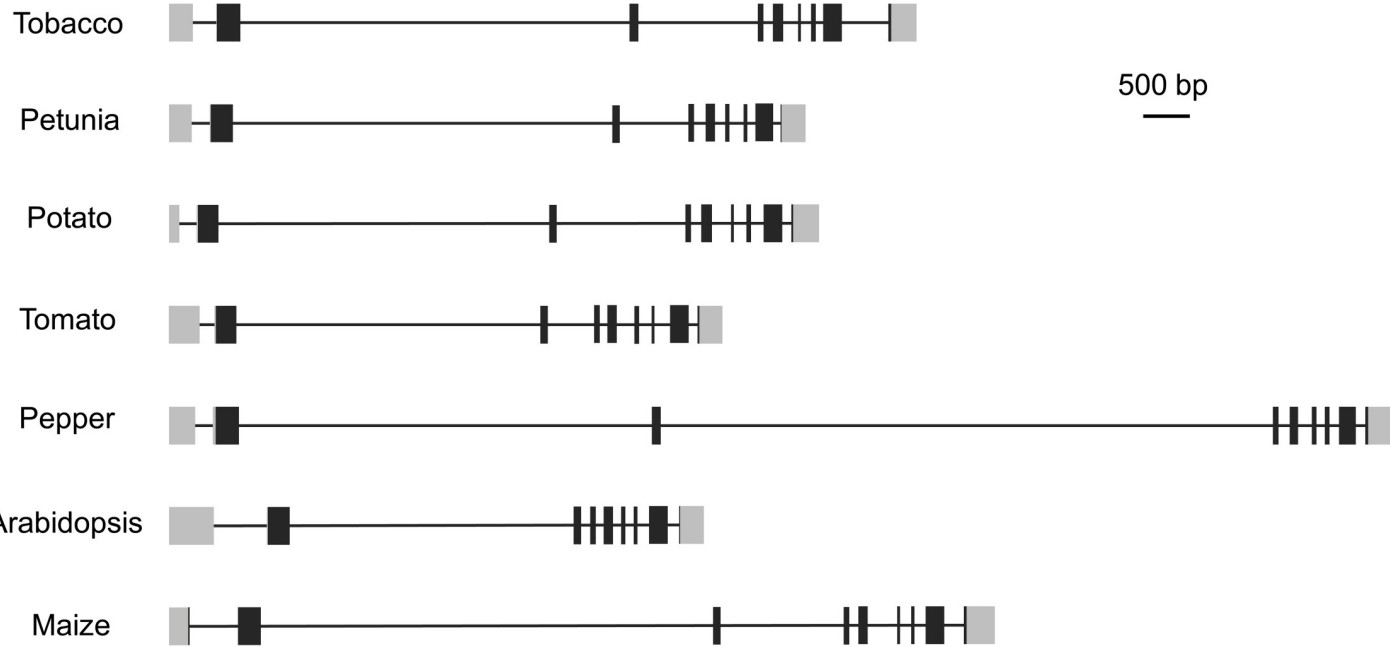

**Fig 7. Structure of *AG* homologous genes in different plant species.** Grey boxes indicate non-coding regions, black boxes represent coding regions, and lines indicate introns.

**Table 1. Comparison of *N.tabacum AG* second intron with those from other plants.**

| Solanaceae species | Percent identity with *N.tabacum* | GenBank accession of *AG* homolog |
|---|---|---|
| *Petunia x hybrida* (petunia) | 69.5 | AB076051.1 |
| *Solanum tuberosum* (potato) | 58.1 | NW_006239189.1 |
| *Solanum lycopersicum* (tomato) | 57.3 | NC_015439.3 |
| *Capsicum annuum* (pepper) | 58.3 | NC_029978.1 |
| Non-Solanaceae species | | |
| *Arabidopsis thaliana* (Brassicaceae) | 48.6 | NC_003075.7 |
| *Zea mays* (maize; Poaceae) | 44.0 | NC_024466.2 |

repressive fragment exists in the *AG* second intron/enhancer that plays crucial role in the repression of vegetative activity of the chimeric promoter by mediating H3K27 tri-methylation. Bioinformatic analysis reveals that the 100-bp repressive fragment has a GAGA factor binding motif, which is conserved among the species of Solanaceae family, underpinning the importance of this element in the vegetative silencing of *AG* gene.

Singer and colleagues [39] reported that 35S enhancer could override the *AGI*-conferred tissue specificity in Arabidopsis, resulting constitutive expression of the AtAGIP controlled gene in the vegetative tissues. We notice that a variant of duplicated enhancer was used in their study. This duplicated 35S enhancer was shown having approximately tenfold higher activity than that of the single 35S enhancer. Furthermore, it could boost the activity of adjacent promoter by several hundredfold [40]. In our study, we used the 351 bp fragment (-396 to -46) of 35S promoter to increase the activity of NtAGIP1. GUS activity detection showed that the activity of NtAGIP1 was increased by one- to two-fold. Meanwhile, the promoter maintains its activity in flower organs. Therefore, it is conceivable that the strong activity of duplicated 35S enhancer is hard to be suppressed by the potential repressive elements of AGI. In other words, the ability of repressive elements of tissue-specific promoters to suppress the global activity of a constitutive enhancer depends on the strength of the enhancer used. Consistently, the 35S promoter altered the level and patterns of activity of adjacent tissue- and organ-specific gene promoters, while the similar constitutive Nos promoter that is weaker than the 35S promoter had no effect on these adjacent promoters [41,42]. Likewise, 35S enhancer could convert the flower-specific promoter PCHS to non-specific gene promoter, while a weaker OCS enhancer was able to increase the activity of PCHS promoter specifically in the targeted tissues [10].

Although weak enhancers can increase activity of tissue-specific promoter, it would not be a good choice when strong expression is needed. Using a strong repressive element or *cis*-repressor may benefit overcoming the constitutive activity of strong enhancer (*e.g.*, 35S) in vegetative tissues. The repressive element of NtAGI-1 we identified is a potential candidate. Du and colleagues used the 163-bp 35S enhancer fragment (-208 to -46) to increase the activity of flower specific CHS promoters that contain vegetative repressive element TACPyAT box. However, the repressive element was unable to suppress the activity of 35S enhancer in vegetative tissues [10,43]. In our study, the vegetative activity of the longer 351-bp 35S enhancer fragment (-396 to -46) that is responsible for the majority of the 35S promoter strength [33] could be suppressed by the repressive element of NtAGI-1, showing a potential of the repressive element in the design of synthetic reproductive promoters.

Adding 35S enhancer can extend the tissue specificity of NtAGIP1 from inner whorls of flower (stamens and carpels) to the outer whorls (sepals and petals), but the activity of 35SNtA-GIP1 does not cross the boundary of flower organs. An array of factors is responsible for the repression of *AG* [44,45]. For example, *APETALA2* is involved in the repressing the expression of *AG* in the outer floral whorls through recruiting the histone deacetylase HDA19, thus

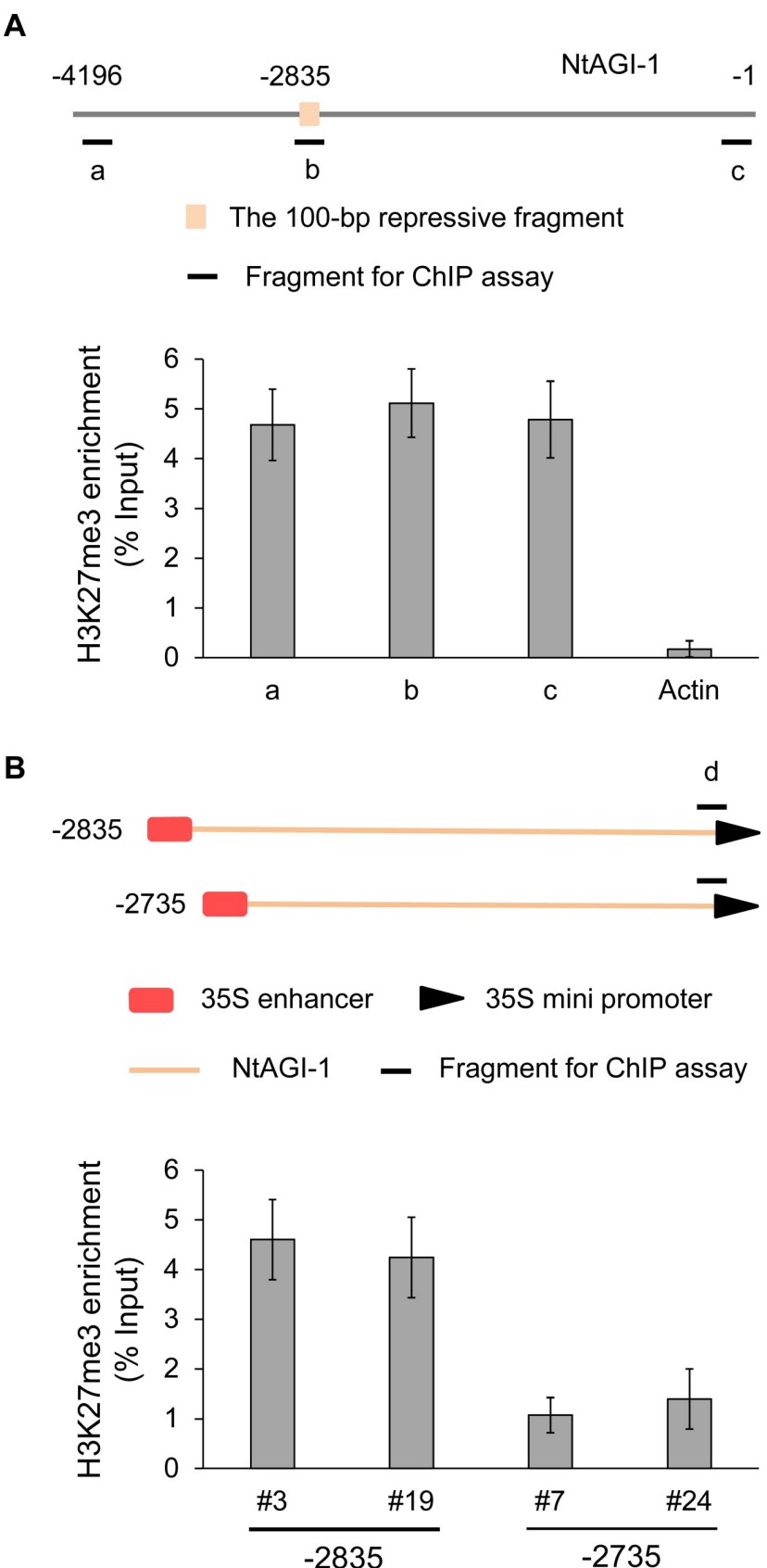

**Fig 8. ChIP assay for H3K27me3 levels in NtAGI-1 intron and chimeric promoters.** (**A**) ChIP assay for H3K27me3 levels at the NtAGI-1 intron in leaves of wild-type tobacco. (**B**) ChIP assay for H3K27me3 levels in leaves of the -2835 and -2735 lines. The genomic fragment from immunoprecipitation for qRT-PCR detection is depicted as black horizontal line. Enrichment was represented as percentage of Input (% Input). Error bars represent the standard deviation of three biological replicates.

preventing gene activation *via* histone deacetylation [23,46]. Polycomb repressive complex 2 (PRC2) is implicate in the repressing *AG* in the vegetative tissues by mediating H3K27 tri-methylation to form epigenetically stable silent chromatin state [37,38]. PRC2 is evolutionarily conserved [47–49]. The recruitment of PRC2 in Arabidopsis relies largely on binding of *trans*-acting factors to *cis* DNA motifs known as polycomb response elements (PREs) [34–36]. Importantly, GAGA factor binding motif has been found to be involved in the recruitment of PRC2 in both Arabidopsis and *Drosophila*. Recently, Wu and colleagues reported that a DNA region within Arabidopsis *AG* intron 2 (+2616 to +3348) is involved in the recruitment of PRC2 to represses *AG* expression in leaves *via* the transcribed noncoding RNA [45]. They pointed out that the identified DNA region also contains PREs which may recruit PRC2. We found that the GAGA factor binding motif is also included in the region of the AtAGI. Bioin-formatic analysis indicates that the GAGA motif is conserved among the Solanaceae family. Furthermore, deletion of the GAGA motif containing 100-bp repressive fragment of NtAGI-1 caused decreased H3K27me3 level and de-repression of the chimeric promoter activity in leaves, suggesting that the GAGA motif may be involved in the recruitment of PRC2 at the *AG* locus for vegetative repression in plants. Thus, our results may inspire studies on the regulation of other floral genes which need to be suppressed so as to allow normal vegetative development [50].

In summary, our work constructs a new chimeric flower-specific promoter for plant engineering, and the 100 bp GAGA motif-containing sequence identified here may provide important information to investigate the regulation of *AGAMOUS* genes in plants.

## Supporting information

**S1 Table. Primer sequences used for plasmid construction and PCR identification of transformants.**
(XLSX)

**S2 Table. Primer sequences used for ChIP.**
(XLSX)

## Acknowledgments

We appreciate the assistance of Dr. Xianbi Li and Dr. Juan Zhao in tobacco transformation and plant management.

## Author Contributions

**Conceptualization:** Ruochen Liu, Yan Pei.

**Data curation:** Ruochen Liu, Yan Pei.

**Formal analysis:** Ruochen Liu.

**Funding acquisition:** Yan Pei.

**Investigation:** Ruochen Liu, Xiuping Zou, You Wang, Qin Long.

**Methodology:** Ruochen Liu, Xiuping Zou, You Wang, Qin Long.

**Project administration:** Yan Pei.

**Resources:** Yan Pei.

**Supervision:** Yan Pei.

**Validation:** Ruochen Liu.

**Visualization:** Ruochen Liu.

**Writing – original draft:** Ruochen Liu, Yan Pei.

**Writing – review & editing:** Yan Pei.

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
