## [Decision Letter · Decision Letter 0]

7 Jan 2020

PONE-D-19-32760

A 100 bp GAGA motif-containing sequence in AGAMOUS second intron is able to suppress the activity of CaMV35S enhancer in vegetative tissues

PLOS ONE

Dear Dr. Yan Pei,

Thank you for submitting your manuscript to PLOS ONE. After careful consideration, we feel that it has merit but does not fully meet PLOS ONE’s publication criteria as it currently stands. Therefore, we invite you to submit a revised version of the manuscript that addresses the points raised during the review process.

We would appreciate receiving your revised manuscript by Feb 21 2020 11:59PM. To enhance the reproducibility of your results, we recommend that if applicable you deposit your laboratory protocols in protocols.io, where a protocol can be assigned its own identifier (DOI) such that it can be cited independently in the future. For instructions see: http://journals.plos.org/plosone/s/submission-guidelines#loc-laboratory-protocols

We look forward to receiving your revised manuscript.

Kind regards,

Serena Aceto, Ph.D.

Academic Editor

PLOS ONE

Journal Requirements:

Reviewers' comments:

Reviewer's Responses to Questions

**Comments to the Author**

1. Is the manuscript technically sound, and do the data support the conclusions?

Reviewer #1: Yes

Reviewer #2: Yes

2. Has the statistical analysis been performed appropriately and rigorously? 

Reviewer #1: Yes

Reviewer #2: Yes

3. Have the authors made all data underlying the findings in their manuscript fully available?

Reviewer #1: Yes

Reviewer #2: Yes

4. Is the manuscript presented in an intelligible fashion and written in standard English?

Reviewer #1: Yes

Reviewer #2: Yes

5. Review Comments to the Author

Reviewer #1: This manuscript by RuochenLiu et al reports repression function of a 100bp element in AGAMOUS 2nd intron due to possible targeting by PPC2 complex. Using this repression function, the authors showed that GUS signal could be restricted to inflorescence although under 35s enhancer. The result is interesting but with current data, it is not publishable on PLOS one. My concerns are:

1. The authors showed that the expression of NtAG intron promoter could be enhanced by 35S due to the presence of the 100bp fragment. However, in figure 5,when this repression fragment alone was under 35S enhancer, there is quite some leaky expression in leaves. This raises one question: for what purpose we could use this repression fragment. For instance, is it possible to use the combination of 35s enhancer, 100bp repression fragment and other flower gene promoter (like AP1,AP3) to enhance the expression of certain flower genes? Would be nice if the author could show some in-planta data, even if it is only for AGAMOUS gene.

2. There are some other strategies to boost gene expression in-vivo, including endogenous promoter driven inducible system or dCas9-based in-vivo activation tool. Compared to the above mentioned methods, usage of 35senhancer has less potential. In this regards, the value of this work is limited.

3. Discovery of the GAGA motif containing fragment is of interest to the community. The author should dig a bit more. Is this motif really bound by PRC2? Could the author at least confirm this region is modified by H3K27me3 and the modification level is dynamic across flower development or at least between leaves and flower? Would be very interesting if the author could delete the 100bp base pair by CRISPR tool and see what the genetic effect is.

4. Besides,the authors should increase the quality of all the figures. They are too blurry.

Reviewer #2: The findings detailed in this manuscript is useful for the plant research community. The experiments and results were appropriately documented overall. However, there are grammatical errors that need to be reviewed and corrected throughout the manuscript. Additionally, in the results section, increase in transformation events with strong GUS expression from 15% to 46% was erroneously mentioned as two-fold increase. Please check the details of the results and figures to avoid these types of errors.

6. PLOS authors have the option to publish the peer review history of their article (what does this mean?). If published, this will include your full peer review and any attached files.

Reviewer #1: No

Reviewer #2: No

---

## [Author Response · Author response to Decision Letter 0]

11 Feb 2020

Response to the editor:

Dear Dr. Serena Aceto,

Thanks for the advice. Accordingly, we improved the manuscript as suggested. The main changes in revised manuscript are as followings:

1. As suggested by reviewers #1, we conducted chromatin immunoprecipitation (ChIP) assay to confirm this region have H3K27me3 modification. We added Figure 8 to show the new results. The procedure for ChIP assay is added in the section of Materials and methods. 

2. The changes are indicated in the revised version labeled 'Revised Manuscript with Track Changes'.

3. The PACE was used to convert figures to meet the figure requirements of PLOS ONE. Thus, all figures submitted before would be replaced with new figures of new format.

4. Please change the support funding as: the Chinese Ministry of Science and Technology of China (Grant 2016YFD0100505).

Response to Reviewers:

Point to point response

Reviewer #1: 

This manuscript by RuochenLiu et al reports repression function of a 100bp element in AGAMOUS 2nd intron due to possible targeting by PPC2 complex. Using this repression function, the authors showed that GUS signal could be restricted to inflorescence although under 35s enhancer. The result is interesting but with current data, it is not publishable on PLOS one. My concerns are:

1. The authors showed that the expression of NtAG intron promoter could be enhanced by 35S due to the presence of the 100bp fragment. However, in figure 5, when this repression fragment alone was under 35S enhancer, there is quite some leaky expression in leaves. This raises one question: for what purpose we could use this repression fragment. For instance, is it possible to use the combination of 35s enhancer, 100bp repression fragment and other flower gene promoter (like AP1,AP3) to enhance the expression of certain flower genes? Would be nice if the author could show some in-planta data, even if it is only for AGAMOUS gene.

Response: We identified the 100bp fragment in NtAG intron that contributes the suppression of activity in vegetative tissues. Sandwiched between 35S enhancer and 35S mini promoter, this 100bp fragment produced a 33% decrease of leaky expression in vegetative tissues compared with the controls (a 100bp fragment from up- or down-stream of the fragment), indicating the suppressive effect of the fragment on vegetative expression. We are uncertain whether the 35S enhancer and 100bp repression fragment be able to enhance the promoter activity for other flower genes. It is a good idea to use the combination of 35s enhancer, 100bp repression fragment and other flower gene promoter to enhance the expression of certain flower genes or test if the fragment is only for AGAMOUS gene or for more genes. We find that when the NtAGI-1sequence is longer than 2835bp no GUS expression was detectable in leaves of all transgenic plants (Figure 4), implying that other sequences/elements in NtAGI-1 are also required to restrict the expression of downstream gene in flower organs. In this study, we focus on the 100bp sequence in the NtAG intron. Next, we are going to identify the sequences (or motifs) which work with the 100bp fragment to suppress the promoter activity in leaf, and then use these sequences together to generate new promoters. We really appreciate the reviewer’s suggestion. 

2. There are some other strategies to boost gene expression in-vivo, including endogenous promoter driven inducible system or dCas9-based in-vivo activation tool. Compared to the above mentioned methods, usage of 35senhancer has less potential. In this regards, the value of this work is limited.

Response: We agree that dCas9-based tool is a powerful tool that can boost the expression of target genes. However, how to accurately regulate the expression of target genes in target organs is still a problem. In the available systems, the chimeric promoter is a simple strategy. Even in CRISPR- Cas9 era, other strategies still have their rooms in plant biotechnology. In addition to promoter design, our study can also provide information to investigate the regulation of AGAMOUS gene.

3. Discovery of the GAGA motif containing fragment is of interest to the community. The author should dig a bit more. Is this motif really bound by PRC2? Could the author at least confirm this region is modified by H3K27me3 and the modification level is dynamic across flower development or at least between leaves and flower? Would be very interesting if the author could delete the 100bp base pair by CRISPR tool and see what the genetic effect is.

Response: Thanks for the suggestion. Accordingly, we conducted chromatin immunoprecipitation (ChIP) assay and showed that in the wild-type tobacco the entire NtAGI-1 intron including the 100bp repressive fragment was enriched for H3K27me3. Meanwhile in the GUS positive -2735 cells, H3K27me3 level of the chimeric promoter were much lower than that in the GUS negative -2835 cells. The results indicate the crucial role of GAGA motif in the vegetative repression of AG and the AGI-derived artificial promoters. We add the new data in Figure 8.

4. Besides, the authors should increase the quality of all the figures. They are too blurry.

Response: This is probably due to the low resolution in submitting. We submit figures with high resolution, hoping that the problem would be resolved.

Reviewer #2: 

The findings detailed in this manuscript is useful for the plant research community. The experiments and results were appropriately documented overall. However, there are grammatical errors that need to be reviewed and corrected throughout the manuscript. Additionally, in the results section, increase in transformation events with strong GUS expression from 15% to 46% was erroneously mentioned as two-fold increase. Please check the details of the results and figures to avoid these types of errors.

Response: Thanks! The error had been corrected. We have carefully edited the manuscript and checked the details of the results and figures.

---

## [Decision Letter · Decision Letter 1]

18 Feb 2020

PONE-D-19-32760R1

A 100 bp GAGA motif-containing sequence in AGAMOUS second intron is able to suppress the activity of CaMV35S enhancer in vegetative tissues

PLOS ONE

Dear Dr. Yan Pei,

Thank you for submitting your manuscript to PLOS ONE. After careful consideration, we feel that it has merit but does not fully meet PLOS ONE’s publication criteria as it currently stands. Therefore, we invite you to submit a revised version of the manuscript that addresses the points raised during the review process.

The manuscript has been improved and now it needs only minor revisions, as suggested by the reviewer. 

We would appreciate receiving your revised manuscript by Apr 03 2020 11:59PM. To enhance the reproducibility of your results, we recommend that if applicable you deposit your laboratory protocols in protocols.io, where a protocol can be assigned its own identifier (DOI) such that it can be cited independently in the future. For instructions see: http://journals.plos.org/plosone/s/submission-guidelines#loc-laboratory-protocols

We look forward to receiving your revised manuscript.

Kind regards,

Serena Aceto, Ph.D.

Academic Editor

PLOS ONE

Reviewers' comments:

Reviewer's Responses to Questions

**Comments to the Author**

1. If the authors have adequately addressed your comments raised in a previous round of review and you feel that this manuscript is now acceptable for publication, you may indicate that here to bypass the “Comments to the Author” section, enter your conflict of interest statement in the “Confidential to Editor” section, and submit your "Accept" recommendation.

Reviewer #1: All comments have been addressed

2. Is the manuscript technically sound, and do the data support the conclusions?

Reviewer #1: Yes

3. Has the statistical analysis been performed appropriately and rigorously? 

Reviewer #1: Yes

4. Have the authors made all data underlying the findings in their manuscript fully available?

Reviewer #1: Yes

5. Is the manuscript presented in an intelligible fashion and written in standard English?

Reviewer #1: Yes

6. Review Comments to the Author

Reviewer #1: I appreciate the efforts by the authors to address my concern. I am satisfied. Could the author integrate some of the responses they gave in the rebuttal into the manuscript? For example, more details about the potential application of the findings.

7. PLOS authors have the option to publish the peer review history of their article (what does this mean?). If published, this will include your full peer review and any attached files.

Reviewer #1: No

---

## [Author Response · Author response to Decision Letter 1]

23 Feb 2020

Point to point response

Reviewer #1: 

I appreciate the efforts by the authors to address my concern. I am satisfied. Could the author integrate some of the responses they gave in the rebuttal into the manuscript? For example, more details about the potential application of the findings.

Response: Thanks a lot for your kind comments and valuable suggestions. We have integrated some information in the reponse into the revised manuscript. In the Introduction of the revised manuscript (Page 2, Paragraph 1, Line 40-44), we explained the merit of the synthetic promoters. In the Discussion of the revised manuscript (Page 9, Paragraph 4, Line 349-351), we summarize the potential application of the findings in the plant biotechnology and in the investigation of AGAMOUS gene in plants.

---

## [Editor Report · Decision Letter 2]

25 Feb 2020

A 100 bp GAGA motif-containing sequence in AGAMOUS second intron is able to suppress the activity of CaMV35S enhancer in vegetative tissues

PONE-D-19-32760R2

Dear Dr. Pei,

We are pleased to inform you that your manuscript has been judged scientifically suitable for publication and will be formally accepted for publication once it complies with all outstanding technical requirements.

With kind regards,

Serena Aceto, Ph.D.

Academic Editor

PLOS ONE

---

## [Editor Report · Acceptance letter]

27 Feb 2020

PONE-D-19-32760R2 

A 100 bp GAGA motif-containing sequence in AGAMOUS second intron is able to suppress the activity of CaMV35S enhancer in vegetative tissues 

Dear Dr. Pei:

I am pleased to inform you that your manuscript has been deemed suitable for publication in PLOS ONE. Congratulations! Your manuscript is now with our production department. 

With kind regards,

on behalf of

Dr Serena Aceto 

Academic Editor

PLOS ONE